# Testosterone Deficiency as One of the Major Endocrine Disorders in Chronic Kidney Disease

**DOI:** 10.3390/nu14163438

**Published:** 2022-08-21

**Authors:** Katarzyna Romejko, Aleksandra Rymarz, Hanna Sadownik, Stanisław Niemczyk

**Affiliations:** Department of Internal Diseases, Nephrology and Dialysis, Military Institute of Medicine, 128 Szaserów Street, 04-141 Warsaw, Poland

**Keywords:** testosterone, chronic kidney disease, hypogonadism, sarcopenia, protein energy wasting

## Abstract

Reduced testosterone concentration is nowadays thought to be one of the main endocrine disorders in chronic kidney disease (CKD). It is caused by the dysfunction of the hypothalamic-pituitary-gonadal axis. The role of testosterone is multifactorial. Testosterone is responsible not only for reproductive processes, but it is a hormone which increases bone and muscle mass, improves lipid profile, insulin sensitivity, erythropoiesis, reduces blood pressure, and ameliorates mood and perception. The implications of hypogonadism in CKD are infertility and loss of libido, reduction of muscle mass and strength, disorders in bone mineralization, the development of sarcopenia and protein energy wasting (PEW), progression of atherosclerosis, increased visceral adiposity, insulin resistance, and anaemia. Reduced testosterone serum concentrations in CKD are associated with increased mortality rate. Testosterone supplementation improves sexual functions, reduces the level of inflammatory markers and blood pressure, stimulates muscle protein synthesis, improves insulin sensitivity and lipid profile, and increases muscle mass, bone mineral density, and haemoglobin concentration. It positively affects mood and well-being. The modes of testosterone supplementation are intramuscular injections, subcutaneous pellets, and percutaneous methods—patches and gels. Successful kidney transplantation may improve gonadal function and testosterone production, however, half of men with low testosterone concentrations before kidney transplantation do not restore hormonal function.

## 1. Introduction

The rapid development of medicine over the last several decades has contributed significantly to the extension of human life, which results in a notable increase in the number of elderly and chronically ill people, including patients with CKD. The incidence of irreversible kidney damage reaches almost 13% of the world population [1]. It becomes important not only to extend the lifetime of patients with CKD, but also to improve its quality. Endocrine disorders associated with CKD including reduced testosterone concentrations take part in the development of cardiovascular complications, hypertension, atherosclerosis, malnutrition, sarcopenia, PEW, anaemia, and lipid hoemostasis disorders, as well as increased insulin resistance [2,3]. Gonadal dysfunction affects almost 50% of men at different stages of renal failure and reaches almost 70% in patients with eGFR lower than 30 mL/min/1.73 m^2^ [4,5]. Low testosterone concentrations are associated with increased all-cause mortality in men with CKD [6].

Changes in sex hormone concentrations in patients with CKD have been neglected for many years. During the last decades, testosterone was thought to be a hormone responsible only for sexual function, reproduction, and libido. However, recent studies have provided information that this hormone affects and is also responsible for normal lipid profile maintenance, correction of body mass composition and fat tissue distribution, reduction of visceral fat mass, increase in bone mineral density and in muscle mass, and improvement of mood and well- being [7,8]. In recent years, the protective role of testosterone in the development of cardiovascular disease has been pointed out more frequently. Testosterone has beneficial effect on blood vessels—decreases blood pressure, reduces vascular calcification and atherosclerosis, maintains positive lipid profile, reduces inflammation processes, and improves insulin sensitivity [9,10]. Low testosterone concentrations seem to be a risk marker for the development of cardiovascular disease [11] and are associated with increased risk of mortality due to cardiovascular and other reasons [11,12,13]. The knowledge of clinical complications of testosterone deficiency in CKD patients seems to be valuable in terms of possible preventive procedures.

## 2. Testosterone Metabolism

The main source of testosterone are interstitial cells of the testes—Leydig cells. In smaller quantities it is synthesized by the adrenal cortex. Testosterone secretion is regulated by luteinizing hormone (LH), while LH secretion is controlled by the hypothalamic gonadotropin-releasing hormone (GnRH), released in a pulsatile manner. There is a negative feedback between testosterone and pituitary gonadotropic cells. Prolactin (PRL) increases the process of steroidogenesis, however, an excess of PRL inhibits testicular function. This may be important for patients with CKD where PRL concentrations are elevated as a result of reduced clearance and increased PRL production [14]. Approximately 0.2% of testosterone is converted to estradiol by aromatase. It is the source of approximately 80% of estradiol circulating in males. The daily synthesis of testosterone in adult males is 3–10 mg, while the reference range of total testosterone (TT) concentration in serum is 280–800 ng/dL (9.7–27.8 nmol/L) [15]. Morning testosterone concentrations are approximately 30% higher than afternoon levels. A total of 70% of testosterone combines with sex-hormone-binding globulin (SHBG), certain part of testosterone combines with albumin, while only 2% circulates as a free fraction which is the most biologically active form.

Testosterone combines with the androgen receptor. The 5α-reductase enzyme converts testosterone into dihydrotestosterone (DHT), which is twice as potent as testosterone itself. The conversion takes place in the prostate, skin, and liver. Testosterone is an anabolic hormone, which increases protein synthesis in target tissues and stimulates cell proliferation and maturation of the tissues. It is converted into inactive metabolites in the liver, kidneys, muscles, and fat tissue.

Testosterone is inactivated by liver oxidases, then combines in hepatocytes with glucuronic acid, and is excreted by kidneys.

Spironolactone and cimetidine are drugs disrupting testosterone synthesis by blocking androgen receptor whereas both spironolactone and ketoconazole additionally reduce 17α-hydroxylase activity. Glucocorticosteroids act directly by gonadal steroid receptor and centrally at the hypothalamic-pituitary level, reducing testosterone synthesis [16]. Tricyclic antidepressants, benzodiazepines, and opiates also reduce testosterone levels by affecting central mechanisms [17]. Treatment of secondary hyperparathyroidism in males with CKD using calcimimetics such as cinacalcet decreases total and free testosterone concentrations [18]. Angiotensin-converting enzyme inhibitors (ACEI), angiotensin receptor blockers (ARB), and statins also interfere with the synthesis of testosterone [19,20].

## 3. Clinical Function of Testosterone

For many years, testosterone was considered to be a hormone taking part mainly in the development of male reproductive organs and tertiary sex characteristics, elongation and thickening of the vocal folds in men, spermatogenesis, erectile function, and libido. Nowadays, it is known that testosterone is involved in various processes occurring in the human body [21]. The knowledge of extensive functions of testosterone and the mechanisms of its actions is crucial to understand the complications of hypogonadism and to implement new therapeutic methods to treat hypogonadism or to prevent its development.

### 3.1. Testosterone and the Lipid Profile

The correct range of serum testosterone protects from atherosclerosis and metabolic syndrome development by reducing the concentration of total cholesterol, low-density lipoprotein cholesterol (LDL-C), and triglycerides (TG) [22] and increasing concentration of high-density lipoprotein cholesterol (HDL-C) at the same time [23]. In males with hypogonadism, testosterone supplementation positively affected lipid profile [24,25]. The prevalence of metabolic syndrome is higher in men with low testosterone concentration in comparison with individuals without hypogonadism [26].

### 3.2. Testosterone and the Insulin Sensitivity

Testosterone is known to improve insulin sensitivity [27]. Low testosterone concentrations are associated with higher insulin resistance [28]. Even in lower normal range, they are related to a higher risk of diabetes development [29]. In patients with hypogonadism, the treatment with testosterone improved insulin sensitivity and reduced glucose and HbA1C concentrations [30]. Moreover, in a large placebo-controlled randomised study of 1007 patients, Wittert et al., observed that testosterone supplementation may prevent the development of diabetes in overweight men [31].

### 3.3. Testosterone and the Cardiovascular System

Testosterone has a protective impact on cardiovascular complications and reduces systolic and diastolic blood pressure [25]. It has been shown that testosterone concentrations are inversely related to blood pressure [32,33]. Testosterone induces vasodilatation by opening potassium channels and by inhibition of L-type calcium channels [34,35]. It also has influence over the synthesis of nitric oxyde (NO) by the activation of endothelial NO synthase (eNOS) [36]. Testosterone receptors are expressed on cardiomyocytes and thus testosterone can modulate intracellular processes of cardiomyocytes. Men with testosterone deficiency are more prone to develop a heart failure in comparison with those without hypogonadism [37]. Furthermore, low testosterone concentrations may have proarrhythmic effects [38] and worsen myocardial perfusion [39]. In patients with coronary artery disease and stable angina, low-dose testosterone implementation reduced myocardial ischemia [40]. The study of Rosano et al., has shown that testosterone levels correlate inversely with the severity of coronary artery disease [41]. Testosterone reduces vascular stiffness and arterial calcification, inhibits the progression of intima—media thickness of the common carotid artery, and also reduces the progression of the atherosclerosis in aorta and carotid arteries [42,43]. A meta-analysis of 43,041 participants proved that low testosterone concentrations are associated with elevated cardiovascular mortality and morbidity [44]. Therefore, it can be assumed that testosterone plays a cardioprotective role [45].

### 3.4. Testosterone and the Body Mass

In aging men, an increase of fat mass and a reduction of lean body mass is observed [46]. In adipose tissue, high aromatase concentrations accelerate the conversion of testosterone to estradiol, which inhibits gonadotropin secretion and thus reduces the synthesis of testosterone [47]. The increase of fat mass is associated with low testosterone concentrations, which is why obese individuals have lower serum testosterone concentrations in comparison with those with correct body weight. Furthermore, adipose tissue is the source of adipocytokines, e.g., leptin. Obese patients have higher leptin concentrations in comparison with normal-weight individuals. Leptin is an anorexigenic hormone which inhibits food intake by stimulation of the satiety centre of the hypothalamus. However, obese individuals develop resistance to leptin. Hyperleptinemia inhibits gonadotropin secretion by the pituitary gland and in consequence suppresses testosterone production [48]. On the other hand, testosterone supplementation in hypogonadal men resulted in a decrease of leptin levels [49,50]. Further studies are required in order to observe if the reduction of leptin levels in obese individuals may positively affect hypogonadism by the rise of testosterone concentrations.

### 3.5. Testosterone and the Immune Processes

Testosterone has an anti-inflammatory function [51,52]. In elderly and obese individuals who have more visceral adipose tissue (which is a major source of proinflammatory cytokines), increased levels of interleukin-6 (IL-6), tumor necrosis factor alpha (TNF-alpha), and interleukin-1β (IL-1β) are associated with a decrease of testosterone concentrations [53]. Testosterone supplementation reduces inflammatory state and lowers the concentrations of inflammatory cytokines [54]. Moreover, testosterone suppresses dendritic cells as well as macrophages and promotes myeloid-derived suppressor cells (MDSCs) and regulatory T cells (Tregs). It also regulates the function of T and B cells by lowering their activation [55]. Page et al., proved that testosterone maintains the balance between autoimmunity and protective immunity by affecting the number of T and CD8^+^ cells, which furthermore suppresses natural killer (NK) cells proliferation [56]. Testosterone concentrations are inversely proportional to plasminogen activator inhibitor (PAI-1) and fibrinogen [57].

### 3.6. Testosterone and the Balance of Bone and Muscle Mass

Testosterone has an anabolic function. In bones after transformation by 5-α-reductase to DHT, it stimulates osteoblasts activity, whereas after the conversion by aromatase to estradiol it inhibits activity of osteoclasts and bone resorption [7,30]. Testosterone suppresses the apoptosis of osteoblasts [58], stimulates the production of interleukin-1β, and ameliorates the mitogenic effect of fibroblast growth factor in osteoblasts [59]. It triggers the differentiation of osteoblasts as well. Low testosterone levels are connected with increased receptor-activator nuclear factor-kappa ligand (RANKL) which elevates the proliferation and activation of osteoclasts. That results in lowering bone density [60], increasing the risk of osteoporosis or osteopenia, and consequently bone fractures [61]. Testosterone deficiency affects 20% of males with symptomatic vertebral fractures and 50% of men with hip fractures [62,63]. The administration of testosterone in males with osteoporosis increased bone mineral density [64,65,66]. Testosterone is also involved in muscle development and in maintaining muscle mass [67]. The administration of testosterone in patients with hypogonadism resulted in an increased lean body mass, muscle mass, muscle strength, and physical performance [10,68].

### 3.7. Testosterone and the Erythropoiesis

Testosterone has a positive impact on the amount of red blood cells and red cell volume, but the mechanisms of the process are not well understood. It stimulates erythropoiesis in several ways: by acting directly on bone marrow stem cells [69], increasing the production of erythropoietin (Epo) [70,71], and inhibiting hepcidin (a protein produced by hepatocytes, which regulates the release and uptake of iron and its transport to erythrocytes) [72]. There are reports, such as the analysis of 5888 men which showed that low testosterone concentrations were significantly associated with anaemia [73]. Coviello et al., proved that the treatment with testosterone results in the rise of haemoglobin and haematocrit [74]. The study by Snyder et al., which included 788 patients with low testosterone concentrations showed that the testosterone supplementation with gel reversed anaemia in around 20% of patients even those with a previous diagnosis and treatment [75]. Further studies are needed to better understand the relationship between hypogonadism and anaemia and thus to prevent the decrease of erythropoiesis.

### 3.8. Testosterone and the Nervous System

Testosterone receptors exist also in the central nervous system. Testosterone positively affects mood, sleep, cognitive activity, emotions, memory, and improves brain flow [76]. Testosterone deficiency results in insomnia, decreased motivation and self-confidence, mood changes, fatigue, and reduced energy. It has been shown that a decrease in testosterone concentrations during the aging process is associated with a higher occurrence of depression [77,78].

## 4. Hypogonadism in Chronic Kidney Disease

Reduced testosterone concentration is one of the main endocrine disorders in men with CKD. It is assumed that decreased testosterone levels occur even in 50% of males with CKD [4]. The cause of hypogonadism in CKD is not fully understood. Both the synthesis of testosterone and its secretion are reduced with a decrease of renal function. In patients with CKD, the ability of testosterone binding and SHBG concentrations are correct, but the concentrations of free and total testosterone are reduced [79]. Decreased testosterone concentrations are due to dysfunction of the hypothalamic-pituitary-gonadal axis. Inappropriate cyclic release of gonadotropin-releasing hormone (GnRH) by the hypothalamus leads to a loss of correct LH release by the pituitary gland and reduced testosterone synthesis [80]. Low testosterone concentrations in CKD may be caused by the resistance to the action of LH at the Leydig cell level [81]. Haemodialysis does not remove testosterone from the circulation; therefore, it cannot be the cause of hypogonadism in dialysis patients [82]. Hypogonadism in CKD may be a result of increased PRL concentrations due to its retention, as well as a three-fold growth in the synthesis of PRL and extension of its half-life [14]. The study of Sahovic et al., showed that the reduction of parathormone (PTH) concentrations in HD patients resulted in the elevation of testosterone levels [83]. Hyperparathyroidism in CKD stimulates PRL synthesis which is the cause of hyperprolactinemia and hypogonadism. The production of testosterone may be impaired by medications frequently used by patients with CKD such as ACEI/ARB, statins, glucocorticosteroids or calcimimetics, e.g., cinacalcet [18,19,20]. Haemodialysis does not normalize testosterone concentrations [2,84].

Serum testosterone concentrations reach the highest level in the morning. The diagnosis of hypogonadism must be based on two blood samples taken in early hours of the day [85]. The study by Cardoso et al., was the first report which proved that the measurements of testosterone concentrations in saliva can be used for the diagnosis of hypogonadism in patients with ESRD [86]. An important conclusion of this study was that the concentration of testosterone in saliva in ESRD patients correlates with the concentration of free fraction of testosterone in plasma [86].

Disorders resulting from testosterone deficiency in CKD may have a significant impact on increased mortality in this group of patients. In the group of men undergoing HD with testosterone concentrations below 6.8 nmol/L, there was a significantly higher mortality rate than in those with testosterone concentrations above 10.1 nmol/L [87]. Gungor et al., proved that the increase of 1 nmol/L testosterone concentration resulted in 7% reduction of mortality rate [87]. In the study of Haring et al., reduced testosterone levels in patients with CKD doubled the risk of all-cause mortality [88]. In dialysis males, low testosterone concentrations are associated with worse prognosis, more frequent cardiovascular events, and higher mortality rate from cardiovascular and other reasons [89].

Symptoms of low testosterone concentrations in patients with CKD are similar to those of hypogonadism during the natural process of aging. CKD is sometimes compared to the state of “accelerated aging”. Decreased libido, erectile dysfunction, inability to achieve orgasm, decreased sexual desire, fatigue, weakness, mood changes, reduced self-confidence, decreased motivation, insomnia, reduced muscle mass and bone mineral density, increased visceral fat mass, and anaemia are observed with the decline of testosterone concentrations [90]. These symptoms in men with CKD occur before the start of dialysis and do not normalize during dialysis therapy. The fastest disorder of hypogonadism is a reduction of energy and libido but interviews with patients with CKD show that impaired sexual function is the main factor that worsens their quality of life.

Low testosterone concentrations in CKD may have a negative impact on many tissues and organ functions. Testosterone deficiency may have an additive negative influence on the development of numerous complications in kidney failure. Moreover, low testosterone concentrations in CKD are associated with increased all-cause and cardiovascular mortality [91]. Cardiovascular complications such as atherosclerosis, hypertension, heart failure, and coronary artery disease are the main cause of death in patients with CKD [92]. Low testosterone concentrations in CKD have a significant contribution to the development of cardiovascular complications. Inflammatory markers such as IL-6, C-reactive protein (CRP), and fibrinogen are elevated in CKD. Low testosterone levels in CKD may intensify inflammatory status. There are studies which reported that low testosterone concentrations in CKD have an inverse correlation with concentrations of inflammatory markers such as IL-6, CRP, and fibrinogen [82,87,91]. Hypogonadism may also play a role in the accelerated process of atherosclerosis in CKD. Low testosterone concentrations are associated with elevated carotid intima-media thickness in CKD patients and also correlate negatively with lipid profile [87,93]. Males treated with HD with low testosterone levels had also a higher risk of the development of atherosclerosis and elevated thickness of intima-media of carotid arteries [94]. In addition to the development of atherosclerosis in men with CKD and hypogonadism, androgen deficiency impairs vasodilatation dependent on blood flow and NO [94]. The study of Kyriazis et al., showed that CKD males with testosterone deficiency had increased stiffness of arteries, which may contribute to the development of cardiovascular disease and increased mortality in these patients [89]. Several studies have shown that reduced testosterone concentrations in CKD correlated inversely with hypertension, endothelial dysfunction, and increased risk of cardiovascular events [87,95]. It was also reported that low testosterone concentrations in CKD intensify the risk of the onset of diabetes [96].

Hypogonadism is an important factor that suppresses mineral bone density and increases the risk of osteoporosis in males [97,98]. Bone mineralization is associated with the appropriate levels of gonadotropins and sex hormones, whereas their reduction results in impaired bone structure and the development of different types of renal osteodystrophy [97,99]. There is an association between serum testosterone and RANKL levels in CKD patients. RANKL is synthesized mainly in bone cells and its concentration reflects the bone turnover. RANKL may mediate the action of testosterone on bone metabolism in patients with CKD. Serum testosterone concentration correlates inversely with RANKL [97]. In CKD low testosterone levels and high RANKL concentrations are associated with elevated bone remodeling [97]. In males with CKD, the degree of decrease in testosterone concentration may be a marker of the structural and mineral disturbances in bones [99].

Protein energy wasting (PEW) was defined in 2007 by the International Society of Renal Nutrition and Metabolism [100]. Decreased body stores of protein and energy in CKD lead to loss of muscle and/or fat mass. Hypercatabolism induced by uremic toxins, malnutrition defined as poor nutrition, and inflammation are the main cause of PEW [101]. Cachexia is the final state of PEW [102]. Sarcopenia is related to poor nutritional status, but it concerns in particular the condition of muscle—the impaired muscle mass, muscle function, and strength. The states that lead to sarcopenia in CKD include increased inflammatory status, insulin resistance, elevated muscle protein degradation, and impaired muscle protein synthesis, as well as low testosterone concentrations [103,104]. Loss of muscle mass leads to frailty which is defined as elevated vulnerability to physical stressors such as illness and physical trauma. It concerns a majority of elderly people but is also associated with renal failure [105,106]. Decreased testosterone level in CKD is one of numerous causes of physical [105]. The presence of sarcopenia is associated with increased mortality in CKD [104,107]. The study of Skiba et al., proved that low testosterone concentrations in CKD are related with lower muscle mass [5]. Rymarz et al., showed that in the group of males undergoing HD low testosterone concentrations were associated with low lean tissue mass [108]. The report by Cobo et al., also showed that males on HD with low testosterone concentrations had reduced muscle mass in comparison with those with correct testosterone levels [109]. Moreover, Kojo et al., concluded that testosterone level may be an independent marker of muscle mass in ESRD patients [110]. PEW and sarcopenia in CKD increase the mortality rate in this group of patients. Hypogonadism is one of the reasons of muscle mass loss in CKD. Therefore, it is crucial to diagnose hypogonadism in men with kidney failure as soon as possible. This may enable to implement testosterone supplementation and to prevent the development of PEW in males with CKD.

Anaemia is one of the main complications of renal failure and the leading cause of anaemia in renal failure is erythropoietin deficiency. Other factors that decrease haemoglobin concentrations in CKD are iron deficiency, imflammation, blood loss, haemolysis, and poor nutritional status [96]. Anaemia in CKD is associated with an increased risk of morbidity and mortality. Testosterone stimulates erythropoiesis, increases the production of erythropoietin, and acts also by the inhibition of hepcidin.

Low testosterone concentrations may be additional cause of anaemia in CKD. Patients with CKD and low testosterone concentrations have more hypochromic erythrocytes and lower levels of erythropoietin [87,111]. In the study of Carrero et al., HD patients with low testosterone had anaemia up to 5.3 times more likely in comparison with those without testosterone insufficiency [112]. In CKD, the correct concentrations of testosterone may contribute to the inhibition of the development of anaemia.

Low testosterone concentrations in CKD are associated with sexual disorders such as decreased libido, erectile dysfunction, and inability to achieve orgasm. All of them lower the quality of life. As it was mentioned before, other disturbances in CKD such as decreased muscle mass and mineral bone density, anaemia, and frailty may be accelerated by reduced testosterone levels. All these derangements may also contribute to the changes of mood and to the development of depression in CKD patients. The diagnosis of depression in CKD may be overestimated and it needs the psychiatric consultation, but it is still significant. Palmer et al., reported that the depression based on interview affected 25% of patients with CKD and in the study of Duan et al., the prevalence of depression was above 22% [113,114]. It is still much more frequent than in the general population where the prevalence of depression is about 7% [115]. The treatment of CKD complications, e.g., hypogonadism, may lead to amelioration of the quality of life and to lowering the incidence of depression in this group of patients.

## 5. Hypogonadism after Renal Transplantation

After kidney transplantation (KTx), gonadal function and testosterone production can return to normal, improve, or stay altered. Earlier studies reported that successful kidney transplantation restores normal function of the hypothalamic-pituitary-gonadal axis, lowers PRL concentrations, and usually normalizes testosterone levels [84,116]. However, contemporary analysis showed that even half of men with hypogonadism before KTx did not restore hormonal function [117]. Interestingly, in the study by Reinhardt et al., in patients after KTx, testosterone levels increased but FSH and LH levels stayed unchanged while prolactin and estrogen levels diminished significantly [117]. However, in other studies LH levels decreased while FSH levels were aberrant [118,119].

Frequency of hypogonadism remaining after transplantation reported in the studies differ from 18 to 50% of grafted men [117,120]. This discrepancy is associated with the fact that the studied groups were heterogenous in terms of age, comorbidities, or dialysis duration. Reinhardt et al., reports higher recovery rate from hypogonadism in patients under 50 years [117]. The remaining hormonal abnormalities after kidney transplantation can be a result of poor graft function, many comorbidities existing also after transplantation, and usage of immunosuppressive therapy. Correlation between testosterone levels and GFR are well known. In the study by Lofaro et al., men with testosterone deficiency presented lower GFR than those with normal testosterone levels [120]. Among comorbidities common in transplanted patients and causing testosterone deficiency are obesity, diabetes mellitus, and hypertension.

Immunosuppressive drugs can influence testosterone production and testicular structure. The effect is dose dependent therefore during the first months after transplantation toxicity of these drugs is strongest. One of the basal groups of immunosuppressant are calcineurin inhibitors (CNI) represented by cyclosporin and tacrolimus, which inhibit interleukin-2 (IL-2) production. Studies based on animal models revealed toxic effect of cyclosporine on testis resulted in decreased testosterone levels and oligospermia [121]. However, clinical studies in humans did not confirm these observations and showed normal semen parameters in men treated with cyclosporine which probably is associated with lower doses of the drugs used in these studies [122,123]. The other CNI is tacrolimus. Animal studies which used relatively high doses of tacrolimus revealed its toxic effects on testis [124]. However, the results were not confirmed in humans. The study comparing hormonal profile in patients after renal transplantation receiving cyclosporine or tacrolimus did not show significant differences in the testosterone, prolactin, and LH and FSH levels between patients treated with these drugs [125].

Another group of immunosuppressants are mammalian target of rapamycin inhibitors (mTOR) such as sirolimus and everolimus which present antifungal, antiproliferative, and immunosuppressive activities. They also can induce hypogonadism. In the study by Tondolo et al., patients after renal transplantation receiving sirolimus had significantly lower testosterone levels and higher LH and FSH levels in comparison to those treated with CNI [126].

On the other hand, improvement of gonadal function after kidney transplantation depends also on the degree of testicular damage before KTx [84,127]. Therefore, sometimes despite the restoration of testosterone production, spermatogenesis can stay altered [118]. Hypogonadism before KTx influence also long-term outcomes. Low serum testosterone level at the moment of KTx is associated with increased risk of graft loss and patient death [128].

## 6. Testosterone Supplementation

According to the British Society for Sexual Medicine (BSSM), International Society for Sexual Medicine and Society for Endocrinology the diagnosis of testosterone deficiency should be based on decreased serum testosterone concentrations and the existing symptoms of hypogonadism [129,130,131]. BSSM recommend testosterone therapy in men with TT concentrations lower than 8 nmol/L or free testosterone (FT) level lower than 180 pmol/L. Therapy with testosterone supplementation is not required for patients with TT concentrations above 12 nmol/L or FT level higher than 0.225 nmol/L [129]. The cut-off for FT is 0.225 nmol/L. ISSM recommend that men with the serum concentrations of TT between 8 and 12 nmol/L may have testosterone deficiency. This group of patients may be offered a testosterone replacement therapy and the improvement of symptoms caused by testosterone deficiency for minimum 6 months of testosterone supplementation confirms the diagnosis of testosterone deficiency [130]. Moreover, the study of Wittert et al., used the value of 14 nmol/L as the cut-off for TT concentrations to prevent the progression of type 2 diabetes. This is the largest Randomized Clinical Trial (RCT) conducted in 1007 patients treated for 2 years with intramuscular injections of testosterone [31].

The goal of testosterone supplementation in CKD is to restore the serum testosterone concentration to normal range and the improvement of complications caused by low testosterone concentrations such as decreased bone mineral density, loss of muscle mass, the development of PEW, anaemia, high inflammatory status, adverse lipid profile, sexual disturbances, and depression. The treatment with testosterone in hypogonadal males with CKD should be taken into consideration in regards with the decrease of mortality rate and the amelioration of the quality of life. The treatment with testosterone is not devoid of side effects and one of them is the accumulation of fluid and swelling. In HD patients the excess of fluid can be removed during HD but in other individuals may be the reason for the discontinuation of treatment. Other side effects which may be present during androgen replacement therapy are weight gain, gynecomastia, polycythemia, and an increased risk of thrombosis, as well as the increase of oral anticoagulants activity and lipid disorders, which paradoxically may aggravate the risk of cardiovascular events [132]. In some groups of patients, exogenous testosterone increased insulin sensitivity, reduced visceral fat mass, the concentration of lipoprotein a (Lpa), fibrinogen, LDL-C, and TG, thus having a beneficial effect on lipid profile. In other groups it increased the concentration of TG and reduced HDL-C, so it might have an adverse effect on cardiovascular complications [133,134]. In the study of Basaria S. et al., with 209 participants, testosterone supplementation was associated with an elevated risk of cardiovascular adverse events [135]. The report of Albert S. et al., with 5328 individuals showed that treatment with testosterone may be associated with increased cardiovascular events especially in the first year of testosterone supplementation [136]. On the other hand, Maggi et al., observed that testosterone treatment, in contrast to age and concomitant cardiovascular disease, was not the predictor of new cardiovascular events [137]. Severe heart failure (class IV according to New York Heart Association), stroke, or myocardial infarction within the last six months are contraindications to testosterone supplementation. Patients with hypogonadism and chronic heart failure may be treated with testosterone but only if regular haematocrit measurements and regular clinical assessment are provided. Cardiovascular risk factors should be assessed in each patient before the beginning of testosterone treatment [138]. Testosterone supplementation may be connected with the rise of haematocrit, therefore testosterone therapy is contraindicated in patients with haematocrit above 54% [138].

The decision of treatment with testosterone should be preceded by exclusion of the presence of prostate cancer by per rectum examination and by measuring the level of prostate-specific antigen (PSA) [90,139]. Testosterone itself does not induce the development of prostate cancer, but it increases the volume of prostate, which may result in the overgrowth of prostate and cause symptoms connected with prostate enlargement [140]. Treatment with testosterone is not recommended in men with palpable nodule or induration and also when PSA concentrations are above 4 ng/mL [141]. Treatment with testosterone should not be initiated in men with PSA level above 3 ng/mL and a high risk of prostate cancer (e.g., African Americans and men with a first-degree relative with diagnosed prostate cancer) [141].

Another possible side effect of testosterone supplementation is worsening of the severity of obstructive sleep apnoea (OSA) [142]. Severe or not treated OSA are contraindications to testosterone treatment [141]. Breast cancer and severe lower urinary tract symptoms such as thrombophilia are the conditions when treatment with testosterone is also not recommended. Men who desire fertility in near term cannot be treated with testosterone as well [141].

Testosterone may be used in the form of intramuscular injections, patches, gel, and subcutaneous pellets [87,143,144]. All of them are in majority well tolerated and efficacious but each form has some limitations and complications. Nasal testosterone has a form of gel, and the absorption of testosterone is through the nasal mucosa. It is a very simple and non-invasive administration with the use of lower dose of testosterone due to sufficient absorption and also avoids the first-pass metabolism. The disadvantages of intranasal treatment may be nostril irritation as it requires several administrations daily. This route of therapy cannot be used in patients with nasal diseases. The patches with testosterone are applied on the back, abdomen, upper arms or thighs. The use of transdermal patches may be limited by skin reactions or the lack of adherence. Although it is also a non-invasive and easy application, patches are nowadays withdrawn in most European countries. Another transdermal route of testosterone supplementation are gel and liquid solutions. The application of testosterone in gel or liquid form should be done on the area of the skin covered by clothes because it minimizes the risk of transfer to women, children or other people who come into contact with the skin of the patient [145]. Gel and patches should be applied once a day. The invasive routes of testosterone administration are subdermal testosterone implants and intramuscular injectable testosterone. The advantages of using testosterone pellet implants are good compliance and the lack of transference. The administration is also much more rare—every 3–6 months. However, tissue incision and local anesthesia are needed to administer the pellet [146]. Similar to patches, pellets are also withdrawn in most European countries. The dose range of intramuscular injections is every 7 to 14 days and infrequent dosage is the advantage of this route. Testosterone injections are connected with fluctuations in serum testosterone levels with high testosterone concentrations during injection and the drop of testosterone level at the end of dosing interval [147]. Testosterone undecanoate (TU) was proved to avoid these fluctuations and it has 15-year follow up in registry studies [148]. TU is a long-acting injectable formulation of testosterone which allows less frequent administrations—every 10 to 14 weeks [149]. One of the complications of injections with testosterone was pulmonary oil microembolism (POME) which was clinically characterized by rough and dyspnea [150]. However, POME is a very rare complication and may be avoided by slow injection. Oral testosterone administration is very convenient, but it requires multiple administrations each day and is associated with fluctuating testosterone levels, therefore this way of treatment was not approved in United States. Buccal patches must be applied two times a day and are well tolerated. Buccal administration may be associated with gum inflammation, gingivitis and unpleasant taste [151]. This form of testosterone supplementation was withdrawn in Europe.

There is little research about the use of testosterone in males with CKD and hypogonadism. The level of testosterone which is necessary for improving sexual functions in males with CKD is the concentration in a lower normal range, while the level of testosterone above these values does not improve sexual functions [152]. The concentration of testosterone must be at the upper limit of normal range, or even above normal range in order to maintain its anabolic function [152]. In 1976 Lim et al., proved that treatment with clomiphene citrate in males with CKD with androgen deficiency resulted in the elevation of testosterone concentrations as well as increased libido and sexual potency [153]. Testosterone supplementation in ESRD resulted in elevation of serum testosterone level and the improvement of sexual functions [154]. In the study of Cangüven et al., with males with ESRD, testosterone supplementation restored testosterone concentrations to normal range and also improved erection and libido [154]. Inoue et al., showed that testosterone replacement therapy in the group of HD hypogonadal men significantly improved the total Aging Males’ Symptoms (AMS) score although there was no significant improvement in subscale scores connected with psychological and sexual functions [140]. In the study of Yeo et al., which included 25 men with CKD and low testosterone levels, the treatment with testosterone was efficacious in moderate and severe CKD. The improvement of the quality of life, the increase of haemoglobin, and grip strength was observed. Testosterone therapy reduced symptoms resulting from hypogonadism in the group of CKD patients [155]. There are several studies which revealed the positive effect of testosterone supplementation on haemoglobin concentrations in men with CKD and hypogonadism [143,156].

The increase in testosterone concentrations in CKD can also be achieved by reducing the concentration of PRL. As a result of decreased levels of PRL, bromocriptine improves sexual function, but its use carries the risk of many side effects [157]. The treatment with lisuride, another dopamine agonist, decreased the level of PRL and increased testosterone concentrations in patients with CKD [158]. Treatment with erythropoietin resulted in an increase of testosterone concentrations and improvement of sexual function in patients with CKD [159]. Low concentrations of zinc in patients with CKD which is the result of appetite disorders, reduced zinc absorption in the gastrointestinal tract, and the loss of zinc during dialysis may also disturb gonadal function. Zinc supplementation in patients with ESRD resulted in the rise of testosterone concentrations, as well as improvement of libido, increased potency, and frequency of sexual intercourse [160].

## 7. Summary

Endocrine disorders are significant complications in patients with CKD and one of them is reduced testosterone concentration. Low testosterone levels in CKD are associated with increased all-cause and cardiovascular mortality rate. The concentrations of testosterone decrease with the reduction of the amount of glomeruli and GFR which results not only in erectile dysfunction, loss of libido, and infertility but also in the reduction of muscle mass and strength, impaired bone mineralization, progression of atherosclerosis, increased inflammation and insulin resistance, hypertension, anaemia, and disorders of adipose tissue distribution. The procedure for hypogonadism treatment in CKD is testosterone supplementation. The routes of its administration are intramuscular injections, subcutaneous pellets and percutaneous methods—patches and gels. The use of testosterone in patients with CKD improves sexual functions, stimulates muscle protein synthesis, increases muscle mass and strength, and improves haemoglobin concentration. Kidney transplantation may improve gonadal function and testosterone production. However, half of men with hypogonadism before kidney transplantation did do not restore hormonal function. This may be the result of poor graft function, comorbidities, the usage of immunosupresive drugs, and it also depends on testicular damage before renal transplantation.

Hypogonadism is nowadays thought to be one of the main endocrine disorders in chronic kidney disease. However, further studies are needed to examine the causes of hypogonadism in chronic kidney disease and the complications caused by low testosterone levels which are known to be numerous in terms of possible preventive and therapeutic procedures.

## Data Availability

The study did not report any data.

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
