# Peer review of "Testosterone Deficiency as One of the Major Endocrine Disorders in Chronic Kidney Disease"

_nutrients, 2022, doi:10.3390/nu14163438_

Round 1
Reviewer 1 Report
In this review the authors reported the main evidence regarding the mechanisms involved in the hypogonadism observed in patients affected by CKD and the few clinical studies focused on the consequences of testosterone deficiency in CKD/ESRD patients.
The review is interesting, mainly because the endocrine disorders in patients with CKD/ESRD, in particular the hypogonadism, are understimated and neglected.
I suggest to improve the manuscript as following reported:
- The authors should lighthen the Section 3 "Clinical function of testosterone" to improve the review readability: It is redundant;
-The authors should include a new section "Hypogonadism in renal transplantation": it is an important topic that the authors have just cited in the Section 4 and that is pertinent to the main topic (CKD). In this new section the authors should report the few studies regarding the recovery of hypogonadism in renal transplant recipients (the authors should cite doi: 10.1007/s40620-018-0513-3).
- The section 5 should belightened (in particular about the Testosterone formulations) and the authors should better discuss the few data emerging from clinical studies which have been conducted among patientes with CKD and renal transplantation.
Author Response
Thank you very much for your careful reading our manuscript and for your comments
Point 1: The authors should lighthen the Section 3 "Clinical function of testosterone" to improve the review readability: It is redundant;
Response 1: We have improved the Section 3 ’’Clinical function of testosterone’’.
Point 2: The authors should include a new section "Hypogonadism in renal transplantation": it is an important topic that the authors have just cited in the Section 4 and that is pertinent to the main topic (CKD). In this new section the authors should report the few studies regarding the recovery of hypogonadism in renal transplant recipients (the authors should cite doi: 10.1007/s40620-018-0513-3).
Response 2: A new section "Hypogonadism in renal transplantation" was included. Some new references were added and the numbers of references were changed.
Point 3: The section 5 should belightened (in particular about the Testosterone formulations) and the authors should better discuss the few data emerging from clinical studies which have been conducted among patientes with CKD and renal transplantation.
Response 3: This section 5 was improved.
Reviewer 2 Report
An interesting paper with lots of useful information. The English require moderate attention and the section on BPH and prostate cancer needs a little work as does the section on therapy as a number of products mentioned are not available and long acting TU (Nebido) is used as frequently as gel in Europe. Some references are very old and thee are better ones as suggested. My suggestions are attached.

Author Response
Thank you very much for your careful reading our manuscript and for your comments
Point 1: An interesting paper with lots of useful information. The English require moderate attention and the section on BPH and prostate cancer needs a little work as does the section on therapy as a number of products mentioned are not available and long acting TU (Nebido) is used as frequently as gel in Europe. Some references are very old and thee are better ones as suggested. My suggestions are attached.
Response 1: We have changed our manuscript according to the attached suggestions.
However, we cannot find the study of Sharma et al Annals of Nephrology 2020 5(1) (verse 370).
In the paragraph starting from ’There is little research...’ (verse 429) we wanted to write about the use of testosterone in males with hypogonadism and chronic kidney disease. We emphasised that there is little research about the use of testosterone in the group of men with chronic kidney disease. That is why we included the study of Lim et al. from 1976 to show that the problem of hypogonadism in chronic kidney disease was noticed long time ago and still it needs new studies to search the causes of low testosterone concentrations in terms of preventive and therapeutic procedures.
Round 2
Reviewer 2 Report
In the abstract enhances perception, otherwise the sentence does not make sense.
The major comment that I have is the 9.7-27.8 nmol/l is a reference range not a normal range. It reflect the levels where 95% of subjects fall and not the levels that are associated with symptoms and require treatment. I think there should be a paragraph quoting guidelines such as BSSM, EAU, or Society for Endocrinology or AUA which suggest treatment below 12 nmol/l. I would also add that the Wittert study used a cut-off of 14 nmol/l to prevent progression to type 2 diabetes and that this is the largest RCT conducted in 1007 patients treated for 2 years. Otherwise readers will conclude that only patients below 9.7 should be treated. Also men with CKD often have raised SHBG and the cut-off for FT is 0.225 nmol/l.
I feel that these points are very important and would add greatly to the practical use of an otherwise very good paper
